# Electroreductive amination of carboxylic acids by cobalt catalysis

Huihua Bi[1], Zhizheng Chen[1], Changsheng Bi[1], Shuanglin Qu [1] ✉ & Jie Liu [1,2] ✉

Catalytic reduction of carboxylic acids to valuable chemicals is highly desirable yet challenging for both biomass conversion and organic synthesis. Here we describe an efficient and sustainable electrocatalytic hydrogenation of carboxylic acids with amines utilizing protons as the hydrogen source. The application of an earth-abundant cobalt complex enables electrochemical generation of a cobalt-hydride intermediate, which serves as the key catalytically active species for this reductive process. Obviating the need for flammable $H_2$ gas or sensitive hydrides, this general hydrogenative coupling of carboxylic acids with amines and nitroarenes allows producing a wide range of structurally diverse complex alkylamines under mild electrocatalytic conditions. Furthermore, the practicality and versatility of this protocol are demonstrated through its application in valuable isotope labeling using readily available deuterium sources.

Direct *N*-alkylation of amines has long-standing interest to synthetic chemists, as the resulting products are highly valuable bulk and fine chemicals such as pharmaceutical agents, agrochemicals, dyes, and natural products[1]. Conventional nucleophilic substitution approach to *N*-alkylated amines mainly relies on the use of alkyl halides as alkylating reagents (Fig. 1a)[2]. However, it suffers from the limitations of stoichiometric amounts of halide waste generation and poor chemoselectivity due to the undesired overalkylations. Although carbonyl reductive amination with aldehydes or ketones has been the benchmark method[3,4], in some cases, their availability, sensitivity, and unwanted side reactions such as aldol condensations still restrict their applications. Carboxylic acids represent an important class of carbonyl compounds that are naturally abundant and structurally diverse from biomass feedstock[5]. The *N*-alkylation of amines using carboxylic acids, a higher-order variant of classical reductive aminations, provides an attractive alternative for the streamlined synthesis of complex alkylamines (Fig. 1b)[6,7]. Pioneered by Gribble and Marchini, the plausibility of reductive alkylation of amines with carboxylic acids was first demonstrated in the 1970s[8,9]. However, the requirement for relatively harsh reaction conditions and superstoichiometric metal hydrides (H⁻) significantly limited the scope of these early methods. Subsequently, a surge of synthetically useful methods have been developed with a diversity of catalytic systems, ranging from the noble metals[10–13] to

base metals[14–17] even main group elements[18–20] in combination with hydrosilanes as terminal reductants (Fig. 1c). More recently, Cole-Hamilton[21], Beller[22,23], and Sundararaju[24] demonstrated the feasibility of this transformation using molecular hydrogen ($H_2$) in the presence of Ru or Co/triphos catalysts. Despite these elegant advances, the utilization of sensitive hydrides or pressurized $H_2$ gas as hydrogen sources might lead to potential security risk, as well as elaborate autoclave manipulation, which restricts their specific utility in synthetic chemistry.

A rising trend in catalytic hydrogenation is the harness electricity as a renewable energy source for green and sustainable synthesis[25–33]. Complementary to classical thermal hydrogenations, electrohydrogenation directly employs protons (H⁺) and electrons (e⁻) as the hydrogen source and redox equivalent to substitute for traditional molecular hydrogen ($H_2$) or hydride (H⁻) donors. Related to this strategy, the electrohydrogenation of various unsaturated bonds, such as carbonyl compounds, has received widespread attention[34–37]. The early examples of electrocatalytic hydrogenation of carbonyl compounds mainly focused on the reduction of relatively high reactive aldehydes and ketones[38–41]. We recently reported selective and efficient electrohydrogenation of less electrophilic nitriles to secondary and tertiary amines catalyzed by a cobalt bipyridine complex[42]. However, to the best of our knowledge, the electroreduction of more

[1]College of Chemistry and Chemical Engineering, State Key Laboratory of Chemo and Biosensing, Hunan University, Changsha, China. [2]Greater Bay Area Institute for Innovation, Hunan University, Guangzhou, China. ✉e-mail: squ@hnu.edu.cn; jieliu@hnu.edu.cn

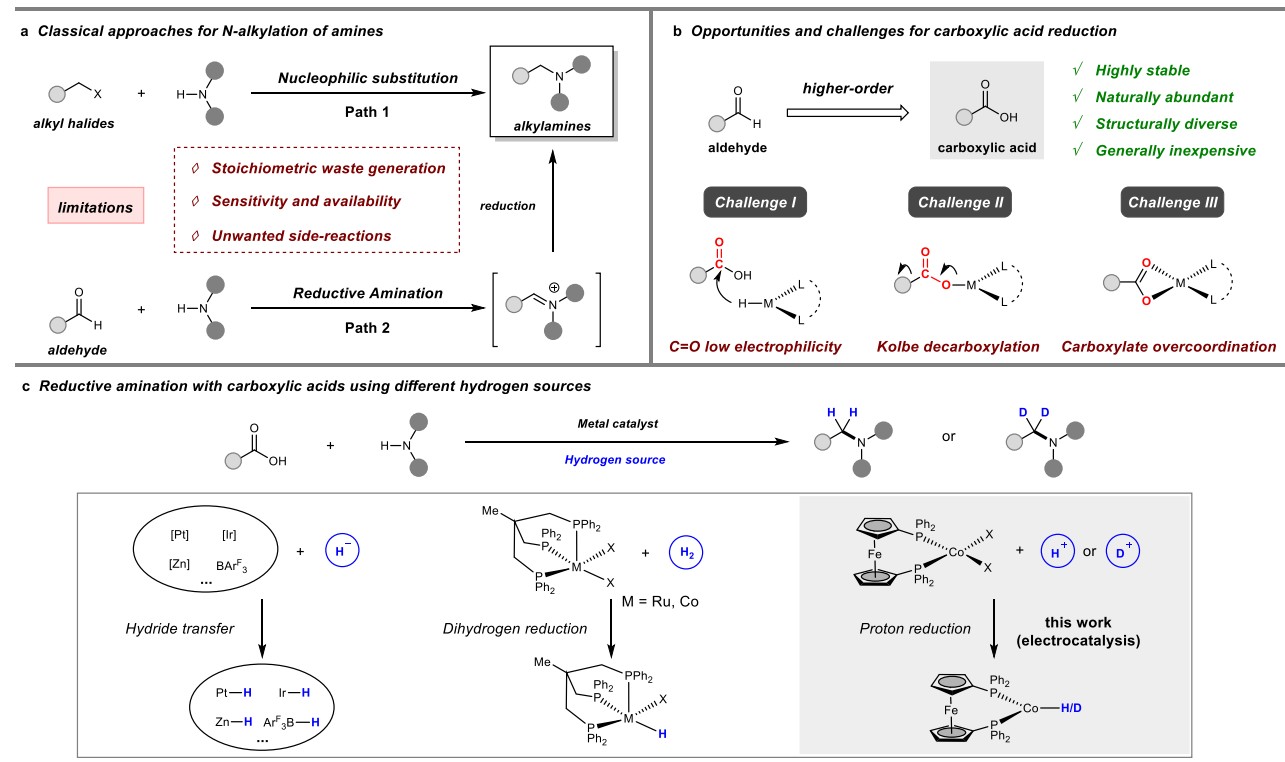

**Fig. 1 | Introduction of *N*-alkylation of amines using different alkylating reagents and hydrogen sources. a** Classical methods for *N*-alkylation of amines. **b** Opportunities and challenges for carboxylic acid reduction. **c** Reductive amination of carboxylic acids using different hydrogen sources.

challenging carboxylic acids has seriously lagged behind related reductions of other carbonyl compounds (Fig. 1b). This is due to the low electrophilicity and thermodynamic stability of the carboxyl group, which renders it less reactive and difficult to reduce, therefore a high negative reductive potential is required[43,44]. Additionally, the Kolbe decarboxylation as a side reaction also competes with the desired carbonyl reduction process[45–47]. Moreover, the interaction of carboxylic acids with the metal catalyst, such as carboxylate over-coordination, may result in detrimental catalyst deactivation[48–50]. Consequently, it is highly desirable to develop a general and robust electrocatalyst system that can reduce carboxylic acids to value-added chemicals, considering their conceivable advantages of natural availability and structurally diversity.

In light of these challenges and opportunities, we herein report a mild and efficient electroreductive amination with carboxylic acids catalyzed by an earth-abundant cobalt complex (Fig. 1c). By utilizing protons as the hydrogen source, this method enables direct *N*-alkylations, including trifluoroethylation and methylation, to access a broad range of complex amines derived from carboxylic acids. Particularly, a key advantage of this electrocatalytic transformation is the feasible and valuable divergent incorporation of deuterium (D) into the resulting amines from readily available deuterated acids.

## Results
### Optimization reaction conditions
At the beginning of this study, a bench-stable, inexpensive, and bulk trifluoroacetic acid (TFA) was selected as the model substrate, as the resulting trifluoroethylation products are highly attractive in medicinally relevant fluorinated building blocks[51]. Compared to volatile trifluoroacetaldehyde or expensive trifluoroethylation reagents, TFA offers a cost-effective and practical alternative for synthesizing valuable β-fluoroalkylamines. Furthermore, as the simplest and ultrashort-chain perfluoroalkyl substance, the efficient and sustainable conversion of TFA into value-added chemicals has garnered significant

attention[52]. To explore this potential, we investigated the reaction parameters for the electrocatalytic hydrogenative coupling of TFA **1** with 4-phenylaniline **2** (Table 1). Based on previous studies[53–55], a key intermediate cobalt-hydride can be generated by cathodic reduction of a cobalt (II) precatalyst and followed by protonation in an acidic medium. Therefore, we evaluated the electrocatalytic performance of cobalt complexes with various ligands. We were pleased to obtain the desired β-fluorinated amine **3** in 93% yield using the commercially available diphosphine dppf (**L1**). Notably, the transformation exhibited good chemoselectivity, as neither reductive defluorination byproducts nor dialkylated species were observed. Other bidentate phosphines such as dppe (**L2**) and binap (**L3**) exhibited moderate reactivity in the model reaction, with the formation of a corresponding amide as a side product. The monodentate as well as tridentate phosphine ligands such as PPh₃ (**L4**) and CH₃C(CH₂PPh₂)₃ (**L5**) also resulted in lower yields. Additionally, bipyridine (**L6**) and other ligands (Supplementary Table 1) did not improve the yield of the *N*-trifluoroethylation product. With the optimal ligand **L1** identified (Table 1, entry 1), we further optimized other reaction parameters. While Co(OTf)₂ provided the best results, other cobalt salts yielded slightly lower efficiencies, and non-noble metal salts such as Fe or Ni failed to produce the desired product (entries 2 and 3). The choice of Lewis acid additive proved critical for effective carbonyl group activation. For instance, ZnCl₂ or BF₃·Et₂O instead of Ti(OⁿBu)₄ did not give better results in this reaction (entry 4). When the reactions were performed at higher or lower temperatures, decreasing yields were observed (entry 5). As to solvent, the use of MeOH or THF suppressed the reaction completely (entry 6). Other sacrificial anodes, such as Mg, Al, or Fe, were inactive under these electrochemically conditions (entry 7). As comparison, stoichiometric reductants like Zn, Mn, Mg dust, PhSiH₃[56], or atmospheric H₂ gas, used in place of electrolysis, showed very low or no catalytic activity in the model reaction (entries 8–10). Control experiments confirmed that the cobalt catalyst was essential, and Ti(OⁿBu)₄ significantly promoted the reaction (entries 11 and 12).

**Table 1 | Effects of reaction parameters[a]**

| Entry | Deviation from standard conditions with L1 ligand | Yield of 3[b] |
|---|---|---|
| 1 | None | 93 |
| 2 | $CoCl_2$, $CoBr_2$ or $Co(OAc)_2$ instead of $Co(OTf)_2$ | 85, 82, 90 |
| 3 | $FeCl_2$ or $NiCl_2$ as a catalyst | 0, 0 |
| 4 | $ZnCl_2$ or $BF_3 \cdot Et_2O$ instead of $Ti(O^nBu)_4$ | 78, 73 |
| 5 | 60 °C or 80 °C instead of 70 °C | 69, 74 |
| 6 | MeOH or THF as solvent | 0, 0 |
| 7 | Mg (+), Al (+) or Fe (+) instead of Zn (+) | 0, 0, 0 |
| 8 | Zn, Mn, or Mg dust instead of electrolysis | 14, 0, 0 |
| 9 | $PhSiH_3$ instead of electrolysis | 0 |
| 10 | 1 bar $H_2$ instead of electrolysis | 0 |
| 11 | No catalyst | 0 |
| 12 | No $Ti(O^nBu)_4$ | 77 |

[a]Reaction condition: trifluoroacetic acid **1** (4.0 mmol), 4-phenylaniline **2** (0.2 mmol), $Co(OTf)_2$ (0.02 mmol, 10 mol%), L1 (0.02 mmol, 10 mol%), $Ti(O^nBu)_4$ (0.2 mmol), MeCN (2.0 mL), toluene (2.0 mL) in an undivided cell with zinc cathode and anode, constant current 20 mA, 70 °C, 3 h, $N_2$.
[b]NMR yield using $CH_2Br_2$ as an internal standard.

## Substrate scope

With the optimal reaction conditions established, we explored the substrate scope of this cobalt-electrocatalytic hydrogenative transformation (Fig. 2). Initially, reductive N-trifluoroethylations using TFA were performed to access a variety of structurally diverse β-fluoroalkylamines (Fig. 2A). In addition to the model substrate 4-phenylaniline, 2-naphthylamine and anilines bearing alkyl, alkoxy, and aryloxy substituents all reacted efficiently, yielding fluorinated amines **4–12** in moderate to good yields (52–83%). A critical issue in hydrogenative reaction is the challenging but highly desirable chemoselectivity of diverse functional groups. Remarkably, anilines containing chlorine, trifluoromethyl, thioether, and sulfonamide functionalities underwent the transformation smoothly, providing synthetically useful yields of products **13–20**. Notably, substrates with free hydroxyl and amino groups were well-tolerated without the need for pre-protection, yielding the corresponding products **21** and **22**. A 2 mmol-scale reaction using 4-(2-aminoethyl)aniline as the substrate afforded product **22** in 63% yield. Under these electrohydrogenative conditions, sensitive and reducible functional groups such as nitrile, ester, and amide were also compatible, and their successful conversion to β-fluoroalkylamines **23–26** further expanded the reaction scope. Additionally, heteroaromatic amines, including quinoline, indazole, benzothiazole, and dibenzofuran, proved to be effective coupling partners, delivering the corresponding products **27–30** in 40–77% yields. Lenalidomide, an immunomodulatory drug used to treat multiple myeloma and anemia, furnished the desired product **31** in 48% yield. Furthermore, diamines reacted smoothly, providing the corresponding dialkylated amines **32** and **33** in 47% and 53% yields, respectively. While alkylamines such as benzylamine or morpholine failed to undergo the desired N-alkylations, this method offered

unique advantages in isotopic labeling. By substituting $CF_3COOD$ instead of $CF_3COOH$[57], we achieved direct deuterium incorporation into β-fluorinated amine **3-d**. This transformation enables the direct introduction of two important functional groups, $CF_3$ and D, to an amine in catalytically one-pot process, highlighting the synthetic utility of this method.

Next, we turned our attention to the scope in terms of other less electrophilic carboxylic acids, including formic acid and aliphatic carboxylic acids (Fig. 2B). Installing a "magic methyl" group into molecules is highly desirable in modern drug discovery and chemical synthesis[58–62]. Among various methylation reagents, formic acid (HCOOH) stands out as one of the most convenient, non-toxic, easily manipulated, and readily available C1 sources, derived from biomass fermentation or $CO_2$ reduction[63,64]. In many cases, the activation and reduction of less electrophilic carboxylic acids often require strong Brønsted or Lewis acid additives[65]. We were pleased to discover that $BF_3 \cdot OEt_2$, rather than $Ti(O^nBu)_4$, served as the optimal Lewis acid additive for the electroreduction of aliphatic carboxylic acids. With this additive, the molecular cobalt-electrocatalyst system enabled efficient N-methylation of diverse amines **34–38** using HCOOH or HCOONa as the methyl source. To further demonstrate the synthetic utility of this methylation method, we applied it to the direct functionalization of several natural products and pharmaceutical molecules. For instance, Cinacalcet **39**, Meclizine **40**, Norquetiapine **41**, and Desloratadine **42** underwent this methylating transformation efficiently, highlighting the compatibility of the electroreductive protocol and its potential utility in late-stage functionalization. In addition to TFA and HCOOH, we also explored electrohydrogenative alkylations with less electrophilic aliphatic carboxylic acids. A representative set of chain and cyclic aliphatic carboxylic acids bearing olefinic and

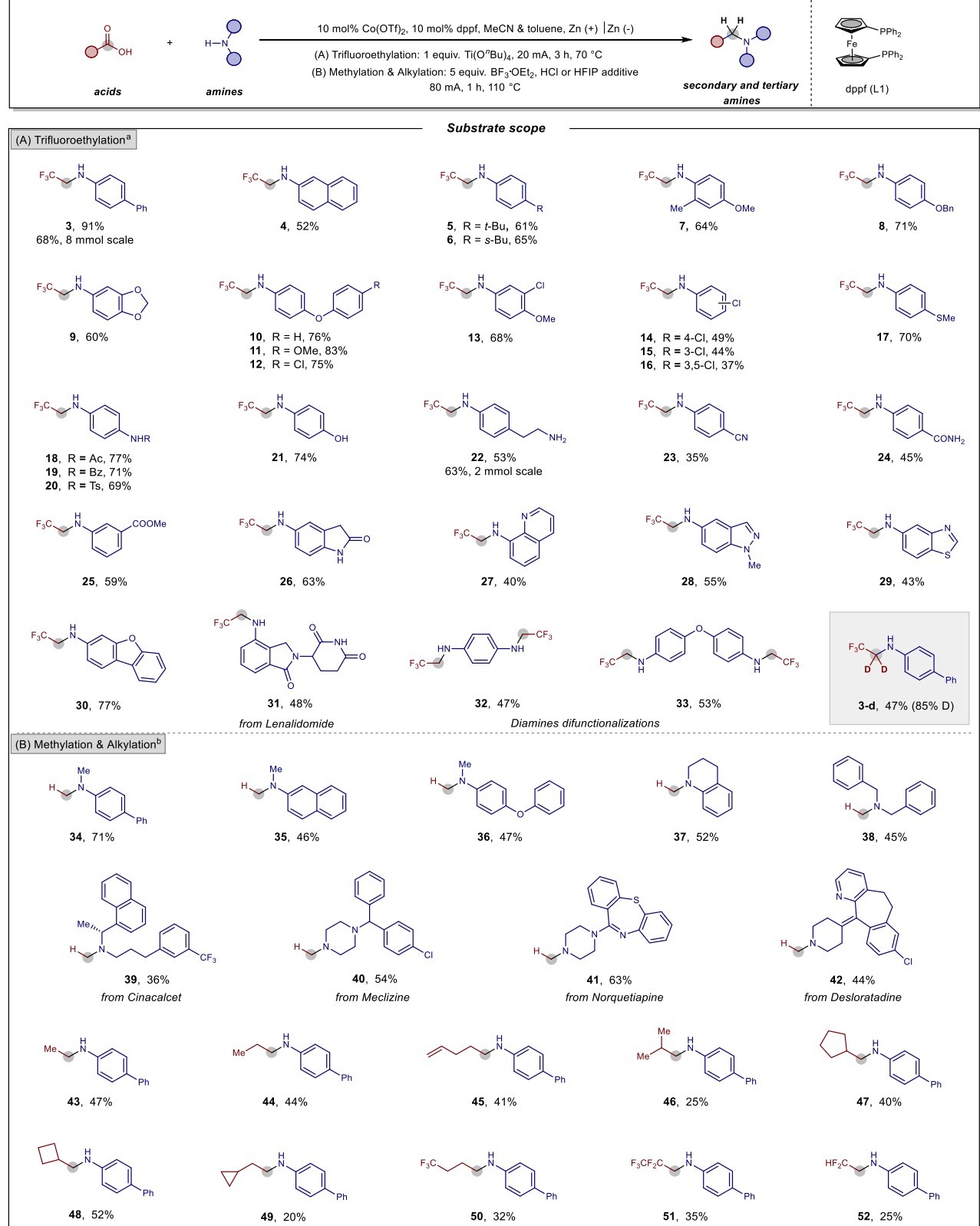

**Fig. 2 | Substrates scope of *N*-trifluoroethylations, methylations, and alkylations of amines with carboxylic acids.** [a] Reaction condition (**A**) as shown in Table 1. [b] Reaction condition (**B**): HCOONa or carboxylic acid (2.0 mmol), amine (0.2 mmol), Co(OTf)₂ (10 mol%), L1 (10 mol%), BF₃·Et₂O (1.0 mmol), HCl (2.4 mmol, for methylation) or HFIP (0.2 mL, for alkylations), MeCN and toluene in an undivided cell with zinc cathode and anode, 80 mA, 110 °C, 1 h, N₂.

**Fig. 3 | Examples of electrocatalytic reductive amination of carboxylic acids with nitroarenes.** [a] 5 equiv. BF$_3$·Et$_2$O instead of Ti(O$^n$Bu)$_4$, HFIP (0.2 mL), 40 mA, 100 °C, 3 h.

perfluoroalkyl functionalities proved compatible with this reductive coupling, yielding the corresponding alkylated anilines **43–52** in moderate yields, whereas substituted quinolines as the side products were observed through the Doebner von Miller pathway (Supplementary Fig. 8). Notably, aromatic acids showed poor conversion under these electroreductive conditions. Despite this limitation, the protocol offers a sustainable alternative to classical Eschweiler-Clarke reaction and reductive amination method, eliminating the need for toxic formaldehyde or unstable aldehydes.

## Synthetic applications

Subsequently, we explored the electroreductive *N*-alkylation of readily available and inexpensive nitroarenes, given that most anilines are derived from their corresponding nitroarene precursors (Fig. 3). This hydrogenative coupling strategy with nitroarenes not only eliminates at least one synthetic step but also leverages bulk and cost-effective starting materials. Using TFA as the carboxylic acid substrate, the cobalt-electrocatalytic trifluoroethylation of various substituted nitroarenes proceeded smoothly, yielding the desired products **53–58**

in 41–69% yields. Notably, pharmaceutical molecules such as Niclosamide **59** and Nimesulide **60** also performed comparably well under these conditions. A variety of reducible or sensitive functional groups in these molecules, including hydroxyl, ether, chlorine, and amide functionalities, were tolerated. Furthermore, aliphatic carboxylic acids, such as propionic acid and cyclobutanecarboxylic acid, were successfully converted into their corresponding *N*-alkylated products **61** and **62**. By eliminating the need for pre-reducing nitroarenes to anilines, this electrochemical approach demonstrates remarkable versatility and practicality, enabling a convenient and economical one-pot synthesis of *N*-alkylamines directly from commercially accessible nitroarenes.

Organic molecules labelled with hydrogen isotopes have garnered considerable attention due to their potential applications in pharmaceutical discovery and radiation chemistry[66–69]. Traditional reductive deuteration processes often rely on expensive and non-recoverable D$_2$ gas or stoichiometric deuteride reagents. Consequently, the development of cost-effective and readily accessible deuterium sources, such as deuterated protons (D$^+$), would be highly desirable and demanding.

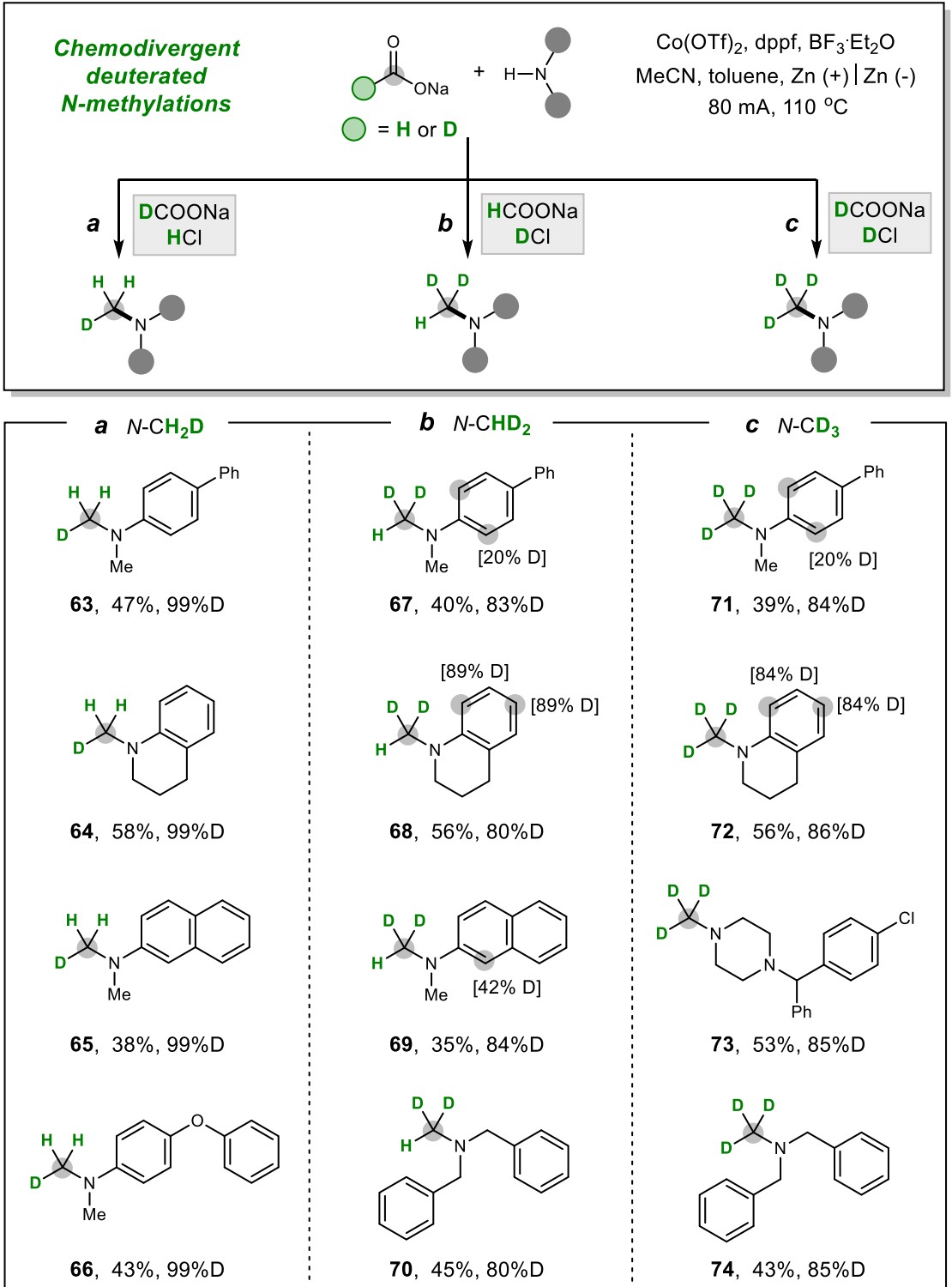

**Fig. 4 | Electrocatalytic chemodivergent deuterated *N*-methylation of amines using isotopic formic acids. a** Installation of *N*-CH₂D groups using DCOONa and HCl. **b** Installation of *N*-CHD₂ groups using HCOONa and DCl. **c** Installation of *N*-CD₃ groups using DCOONa and DCl.

In this work, we present an electrocatalytic approach for the incorporation of deuterium (D) at the α-positions of amines (Fig. 4). Remarkably, we achieved a practical and chemodivergent deuteration of *N*-methylation by strategically tuning the isotopic formates (HCOONa or DCOONa) and protic acids (HCl or DCl) as hydrogen (H) or deuterium (D) sources. Using this method, three distinct deuterium-labeled amines (−CH₂D, −CHD₂, −CD₃) were obtained in a selective

manner. For instance, the d¹-methylated amines **63**−**66** were efficiently synthesized with high deuterium incorporation by employing DCOONa as the C1 source and aqueous HCl as the proton donor. When commercially available DCl was used, the valuable d²-methylation of amines with HCOONa was achieved, yielding products **67**−**70** with over 80% deuterium incorporation. Furthermore, the combination of DCOONa with DCl enabled full d³-methylation of *N*-methyl-4-

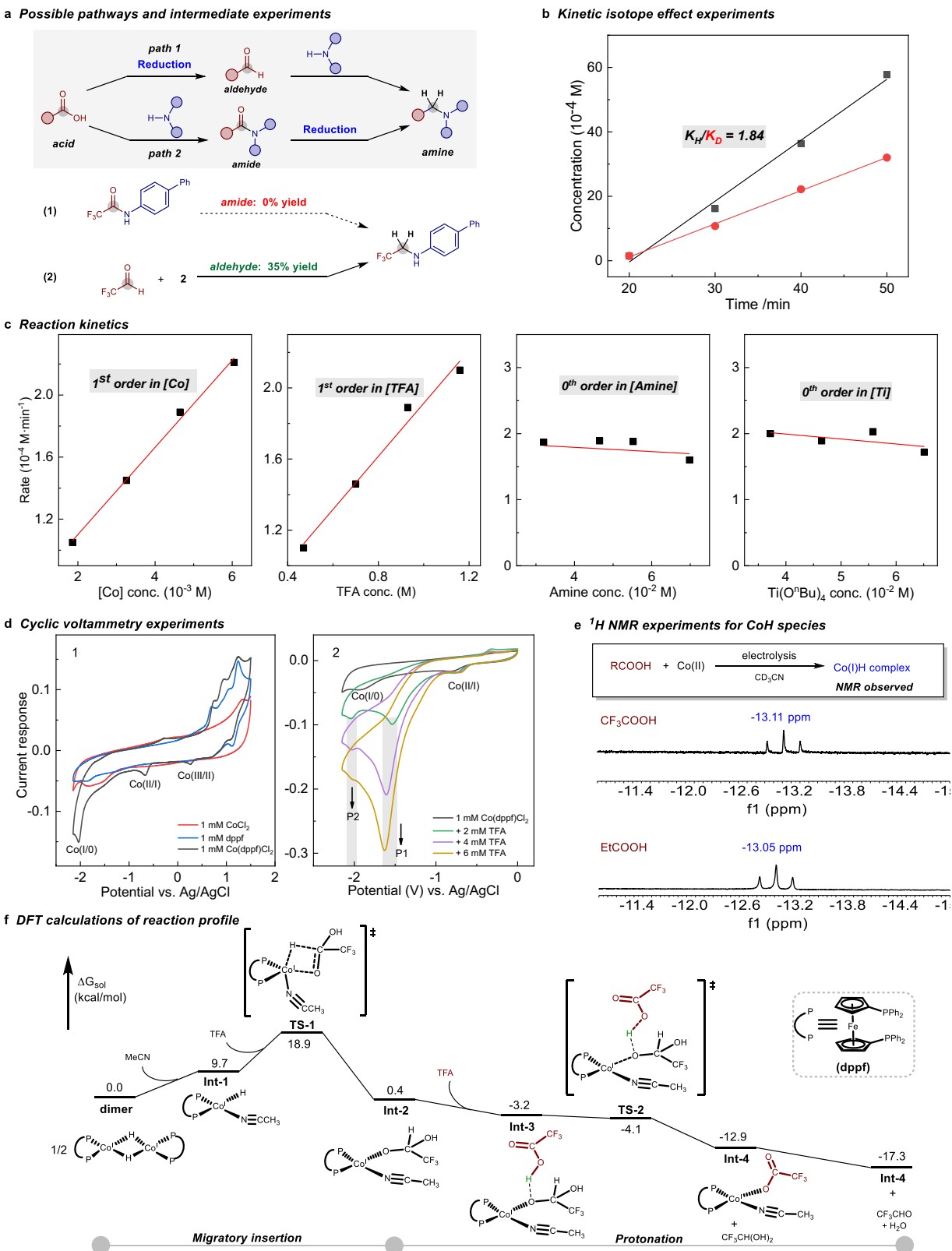

**Fig. 5 | Mechanistic investigations. a** Possible pathways and intermediate experiments. **b** Kinetic isotope effect experiments. **c** Reaction kinetics. **d** Cyclic voltammetry experiments. **e** $^1$H NMR experiments for CoH species. **f** DFT calculations of reaction profile.

phenylaniline **71**, tetrahydroquinoline **72**, meclizine **73**, and dibenzy-lamine **74**. This divergent incorporation of deuterated magic methyl groups represents a powerful tool for isotope labeling of small molecules, offering substantial potential for drug optimization and development.

## Mechanistic investigations

To gain some insights into this electroreduction of carboxylic acids, a series mechanistic experiments were conducted. As shown in Fig. 5a, two plausible pathways were proposed for the hydrogenative coupling of carboxylic acids with amines. The carboxylic acid can be reduced to

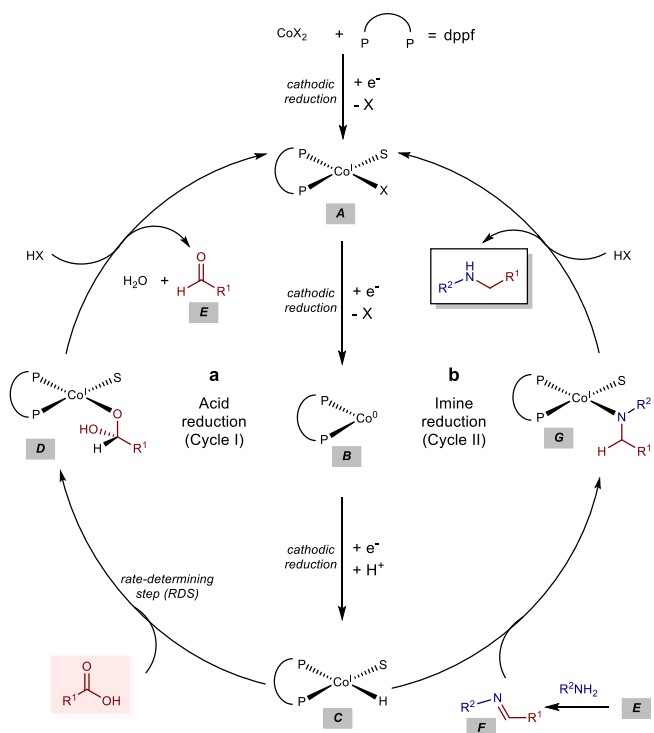

**Fig. 6 | Proposed mechanism. a** Acid reduction (Cycle I) **b** Imine reduction (Cycle II).

an aldehyde, which subsequently undergoes reductive amination to form the desired alkylamine. In addition, the amide formation followed by deoxygenative reduction might also exist as an alternative pathway. To elucidate the mechanism, we investigated the potential intermediacy of amide or aldehyde in this electroreduction. When the reaction was conducted with amides under electrocatalytic conditions, the expected corresponding amines were not formed. In contrast, treatment of trifluoroacetaldehyde hydrate as the alkylating reagent in place of TFA, the desired N-trifluoroethylative product was obtained in 35% yield. These findings suggest that amide formation prior to reduction is not involved in this transformation.

Next, we conducted further kinetic studies to elucidate the rate-determining step of the carboxylic acid electrohydrogenation process. Initially, kinetic isotope effect (KIE) experiments were performed using deuterated and non-deuterated TFA as substrates (Fig. 5b). The rate profiles from the KIE experiments revealed a $K_H/K_D$ value of 1.84, indicating a primary KIE and suggesting that hydrogen transfer is a key step in the reaction mechanism. Subsequently, we investigated the reaction kinetics by varying the concentrations of the cobalt catalyst, TFA, 4-tert-butylaniline, and Ti(O$^n$Bu)$_4$ within the first 50 min of the reaction (Fig. 5c). The kinetic analysis demonstrated a first-order dependence on the concentrations of both the cobalt catalyst and TFA, while the reaction exhibited zero-order dependence on the concentrations of 4-tert-butylaniline and Ti(O$^n$Bu)$_4$. Although Ti(O$^n$Bu)$_4$ is not involved in the rate-determining step, its role as a Lewis acid in activating the carboxylic acid carbonyl group was confirmed by [13]C NMR spectroscopy (Supplementary Fig. 30). These findings suggest that the reduction of the carboxylic acid by the cobalt catalyst is involved in the turnover-limiting step.

Moreover, cyclic voltammetry (CV) experiments were investigated to acquire further understanding for this transformation. As shown in Fig. 5d, measurement of CV curve for the ligand-free cobalt catalyst showed weak redox peaks in a wide range scan from +1.5 to −2.2 V vs. Ag/AgCl (red line). Notably, the application of a dppf-ligated cobalt complex displayed three distinct cathodic peaks at +0.3, −0.7,

and −2.0 V vs. Ag/AgCl, corresponding to the Co(III/II), Co(II/I), and Co(I/0) redox processes, respectively (black line). To further probe the interaction between the cobalt complex and TFA, CV measurements were performed with the addition of 2–6 mM TFA to the cobalt complex solution. This resulted in the disappearance of the Co(I/0) peak and the emergence of a new potential (P1) at −1.5 V vs. Ag/AgCl, accompanied by an enhanced current response. This observation suggests the coordination of trifluoroacetate to the cobalt complex. Additionally, a more negative peak (P2) at −2.1 V vs. Ag/AgCl was detected, indicating the possible formation of a more reductive cobalt species (Co-H) following acid addition[70]. To further characterize this key intermediate, NMR experiments were performed. The reactions were carried out by direct electrolysis of carboxylic acids (TFA or propionic acid) with the cobalt complex in CD$_3$CN for 1.5 h. As shown in Fig. 5e, the [1]H NMR spectra exhibited similar chemical shifts at approximately −13.1 ppm, which can be attributed to the hydride signal of a Co(I)-H species[71,72]. Furthermore, the corresponding aldehyde and alcohol products resulting from acid reduction were also detected (Supplementary Fig. 29). These findings provide evidence for the generation of an active cobalt(I) hydride intermediate under electrocatalytic conditions, which plays a pivotal role in the hydrogenation of carboxylic acids.

Furthermore, we employed density functional theory (DFT) calculations to investigate mechanistic details of the key step of hydride transfer to carboxylic acid (Fig. 5f and Supplementary Data 1). Based on the experimental results, the active species is identified as a cobalt(I) hydride complex. Previous studies have shown that cobalt(I) hydride complexes typically form dimeric structures[73,74], which are considered the resting state and thus serve as the starting point for the energy profile in our DFT calculations. Initially, the Co(I) dimer transforms into the active Co(I)-H species (**Int-1**) with the assistance of solvents, which is endergonic by 9.7 kcal/mol. Next, the Co(I) hydride undergoes migratory insertion into TFA, proceeding through the transition state **TS-1** to form a hemiacetal complex, **Int-2**. This hydride transfer step exhibits an overall activation barrier of 18.9 kcal/mol, representing the rate-determining step of the reaction[75]. Following this, the protonation process occurs. The proton of TFA initially approaches **Int-2**, forming a hydrogen bond with the hemiacetal oxygen atom, resulting in **Int-3**, with a decrease in energy by 3.6 kcal/mol. Subsequently, the proton transfer proceeds readily via the transition state **TS-2**, leading to the formation of 2,2,2-trifluoroethane-1,1-diol and a carbonate complex **Int-4**. This protonation process is energetically favorable, with a decrease of 13.3 kcal/mol from **Int-2** to **Int-4**. The transition state **TS-2**, while identifiable in terms of electronic energy in the gas phase, disappears after corrections for solvation effects and thermal contributions, indicating that the proton transfer can proceed smoothly. The resulting 2,2,2-trifluoroethane-1,1-diol is unstable and readily converts to the more stable trifluoroacetaldehyde and H$_2$O, which is energetically downhill by 4.4 kcal/mol. The subsequent reaction processes and competitive reaction pathways have also been calculated and are presented in the Supplementary Figs. 31–36.

Based on these results, we propose a plausible mechanism in Fig. 6. Firstly, the monoligated cobalt complex is reduced to Co(I) **A** and then to Co(0) **B** species through a stepwise electron reduction process on cathode. In the presence of an acid, this complex **B** is further converted to the corresponding Co(I) hydride complex **C**, which is the key active species to initiate the following catalytic cycles. In cycle I, which involves the reduction of the carboxylic acid, the electrogenerated Co(I)−H species **C** transfers a hydride to the carboxylic group, forming a hemiacetal intermediate **D**. This hydride transfer step has been identified as the rate-determining step of the reaction. Following this, intermediate **D** undergoes rapid protonation by an acid, releasing the aldehyde **E** and regenerating the Co(I) species **A**, thereby completing cycle I. In cycle II, the Co(I)−H species **C** reacts with the imine **F**, which is generated through the rapid condensation of

the aldehyde **E** with the added amine, to form the cobalt complex **G**. Finally, protonation of complex **G** yields the desired amine product and regenerates the Co(I) species **A**, thus completing cycle II.

## Discussion

In summary, we have developed a selective and efficient cobalt-electrocatalytic reductive amination method using stable and versatile carboxylic acids as substrates. Mechanistic studies reveal that an electrochemically generated Co(I)−H species serves as the key intermediate for carboxylic acid reduction, with hydride transfer to the carboxylic acid identified as the rate-determining step. By utilizing readily available protons as the hydrogen source, this electrochemical approach offers a mild, safe, and easily manipulated method for synthesizing complex alkylamines with a broad range of functionally and structurally diverse substituents. Furthermore, this protocol is applicable for the practical and cost-effective incorporation of deuterium isotopes into various amines, including pharmaceutical molecules. We anticipate that this transformation will enable robust and sustainable access to value-added alkylamines, providing a viable alternative to current benchmark methods.

## Methods

### Procedure for the synthesis of compound 3

A 10 mL glass tube equipped with a magnetic stir bar was charged with 4-phenylaniline **2** (0.2 mmol), Co(OTf)$_2$ (0.02 mmol), and dppf (0.02 mmol). Toluene (2.0 mL), acetonitrile (2.0 mL), Ti(O$^n$Bu)$_4$ (0.2 mmol), and TFA **1** (4.0 mmol) were then added sequentially. The reactor was equipped with Zn plates as both the cathode and anode (size 2.5 cm × 1.0 cm × 0.05 cm). The reaction was purged with N$_2$ for three minutes, and the tube was wrapped with tape. The reaction was then electrolyzed under a constant current of 20 mA for 3 h at 70 °C. Upon completion of the reaction, the mixture was quenched with saturated NaHCO$_3$ solution and diluted with ethyl acetate. The organic layer was washed with saturated NaHCO$_3$, dried over anhydrous Na$_2$SO$_4$, and concentrated under reduced pressure. The resulting residue was purified by silica gel flash chromatography to afford the product **3** in 91% isolated yield.

## Data availability

The data reported in this paper are available within the article and its Supplementary Information files. All data is also available from the corresponding author upon request.

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

## Acknowledgements

The authors are grateful for the financial support from the National Natural Science Foundation of China (22301073, J.L.), the Science and Technology Innovation Program of Hunan Province (2021RC3053, S.Q. and 2021RC3056, J.L.), and the Fundamental Research Funds for the Central Universities. The authors also thank the Analytical Instrumentation Center of Hunan University for mass spectrometry analysis.

## Author contributions

H.B., S.Q., and J.L. conceived and designed the project. H.B. and C.B. conducted the experimental part. Z.C. conducted the theoretical calculations. S.Q. and J.L. wrote the manuscript. All authors contributed to analyzing the data and editing the manuscript.

## Competing interests

The authors declare no competing interests.
