## [Transparent Peer Review file · Nature Communications]

Electroreductive amination of carboxylic acids by cobalt catalysis

Corresponding Author: Dr Jie Liu

Version 0:

Reviewer comments:

Reviewer #1

(Remarks to the Author)

Liu and Qu developed an electrochemical strategy for reductive aminations using stable carboxylic acids, leveraging electrons and protons as redox equivalents and hydrogen sources. This approach offers a sustainable route to diverse alkylamines and highlights potential applications in deuterium labeling. The authors also provided evidence for this mechanism through kinetic studies as well as theoretical calculations. A cobalt-hydride species was proposed and identified as the key intermediate in this transformation. While this study presents a well-structured approach and makes a valuable contribution to the field, certain aspects would benefit from additional clarification and investigation to strengthen the manuscript prior to publication:

1. The kinetics presented in Fig. 5c are crucial for elucidating the reaction mechanism. Has the author studied the influence of current on the reaction process, particularly how it affects the kinetic reaction order of the substrate and catalyst? Could variations in current density lead to a change in the rate-determining step?
2. In terms of the reaction pathway, while the outer-sphere mechanism (e.g., TS-2 in Fig. 5f) is compelling, have the authors explored the possibility of an inner-sphere mechanism to rule out alternative pathways?
3. In Fig. 5f, a potential intramolecular 1,3-H shift from intermediate Int-2 may also exist as an alternative route to liberate trifluoroacetaldehyde while generating the Co(I)-OH species. This process should be discussed, as this alternative route might offer additional mechanistic insights.
4. Did the author observe any defluorinative side-reactions with trifluoroacetic acid during the electrocatalytic process in Table 1?
5. Compared to trifluoroacetic acid, significantly lower yields with simple alkyl carboxylic acids were observed in Fig. 2. An explanation for these results is required.
6. When primary amines were used as substrates, was there any detection of dialkylated products? Clarifying this would provide insight into the reaction's selectivity.
7. Regarding the role of Lewis acids such as Ti(OBu)₄ and BF₃·OEt₂ in activating carboxylic acids, could experimental or computational evidence be provided to substantiate the proposed mechanism?

Reviewer #2

(Remarks to the Author)

In the titled manuscript, Liu and coworkers reported an interesting electrocatalytic reductive amination of carboxylic acids via cobalt catalysis. This efficient and sustainable method utilizes a key cobalt-hydride intermediate to facilitate the hydrogenation of carboxylic acids with amines, providing a valuable approach for synthesizing various amines and deuterated derivatives. The products are well-characterized, ensuring reproducibility for researchers interested in amine synthesis. Furthermore, control experiments, kinetic isotope effect studies, and computational analyses support the proposed reaction mechanism. Therefore, this manuscript could be published in Nature Commun, provided that the following points are sufficiently addressed.

- 1, In terms of carboxylic acids scope, trifluoroacetic acid demonstrated high efficiency, achieving a 93% yield of product. However, simple aliphatic carboxylic acids (e.g., acetic acid, propionic acid and others) provided much lower reactivities. Could the authors explain the poor performance of aliphatic carboxylic acids in this system, including any observed side reactions? What happens when aromatic carboxylic acids are used in this reductive reaction?
- 2, Are simple amino acids compatible with this electrochemical system, including protected or unprotected amino acids?

- 3, Regarding the amine substrates, how do alkylamines react with trifluoroacetic acid under electrocatalytic conditions, particularly considering they form products when formic acid is used as the reagent? Furthermore, would this electrochemical system be suitable for processing substituted hydroxylamine derivatives?
- 4, The function of $\text{Ti}(\text{OnBu})_4$ as an additive in this electrocatalytic reduction needs to be elucidated.
- 5, What are the possible reaction intermediates when using nitroarenes as substrates? A plausible reaction pathway is required.
- 6, Given that the authors identified aldehydes as reduction intermediates in their mechanistic studies, has the possibility of imine intermediates been explored? Were any imine species detected under the reaction conditions?
- 7, Please carefully check the tables, figures and main text, and unify the format of the reaction equations, e.g. some type errors in Fig. 2 and Fig.4.

Reviewer #3

(Remarks to the Author)

Liu, Qu and co-workers developed an electrocatalytic reductive amination strategy that achieved hydrogenative coupling between carboxylic acid compounds and amines/nitroarenes. This research employed protons as the hydrogen source and a cobalt complex as the catalyst, which in situ generated active cobalt-hydride catalytic species under electrochemical conditions, thereby enabling efficient construction of alkylamine compounds. Through controlled experiments including cyclic voltammetry analyses and ^1H NMR experiments, the authors monitored and characterized the $\text{Co}(\text{I})\text{-H}$ species. Based on controlled experiments including amide/aldehyde additive comparisons and kinetic isotope effect studies, combined with DFT calculations, the authors proposed a reaction mechanism involving the reduction of carboxylic acids to aldehydes followed by reductive amination to construct alkylamine compounds. In conclusion, this study on the selective cobalt electrocatalytic reductive amination of carboxylic acids using protons as the hydrogen source demonstrates sufficient novelty in the reaction design and a thorough exploration of the reaction mechanism. This reviewer believes the manuscript can be published in Nature Communications after a minor reversion.

1. In Fig. 5f, this reviewer suggests that the alternative pathway involving direct $\beta\text{-O}$ elimination of Int-2 to generate a $\text{Co}(\text{I})\text{-OH}$ complex and trifluoroacetaldehyde simultaneously could be considered.
2. In the Computational Methods section, please provide explicit details regarding the parameter setting of mixed-solvent between Toluene and Acetonitrile to ensure reproducibility and facilitate independent validation by readers.
3. With reference to Fig. 5f of the main text and theoretical study on $\text{Co}\text{-H}$ catalyzed hydrogenation of ketones (Catal. Sci. Technol., 2019, 9, 5315-5321), the three-coordinate alkylamine cobalt(I) complex generated by migratory insertion of imine into the $\text{Co}(\text{I})\text{-H}$ bond in Fig. S25 may be unstable and coordinatively unsaturated. Therefore, it is recommended to consider appending another ligand (e.g., acetonitrile) to stabilize the intermediates Int-11 and to promote the subsequent protonation step.

Version 1:

Reviewer comments:

Reviewer #1

(Remarks to the Author)

The authors have thoroughly addressed all of this reviewer's concerns, and the manuscript has been significantly refined. Congratulations to the authors! This reviewer recommends publication in Nature Communications.

Reviewer #2

(Remarks to the Author)

The authors have fully addressed my comments. Publication as it is recommended.

Reviewer #3

(Remarks to the Author)

The revised manuscript now addresses this reviewer's concerns adequately and is suitable for publication.

Response letter

Reviewer #1 (Remarks to the Author):

Liu and Qu developed an electrochemical strategy for reductive aminations using stable carboxylic acids, leveraging electrons and protons as redox equivalents and hydrogen sources. This approach offers a sustainable route to diverse alkylamines and highlights potential applications in deuterium labeling. The authors also provided evidence for this mechanism through kinetic studies as well as theoretical calculations. A cobalt-hydride species was proposed and identified as the key intermediate in this transformation. While this study presents a well-structured approach and makes a valuable contribution to the field, certain aspects would benefit from additional clarification and investigation to strengthen the manuscript prior to publication:

1. The kinetics presented in Fig. 5c are crucial for elucidating the reaction mechanism. Has the author studied the influence of current on the reaction process, particularly how it affects the kinetic reaction order of the substrate and catalyst? Could variations in current density lead to a change in the rate-determining step?

Our response: We examined the effect of varying electric currents on the reaction. The kinetic reaction orders of TFA and the cobalt catalyst were determined at current of 15 mA and 10 mA, with the results presented below.

(1) Kinetic reaction orders of TFA and cobalt catalyst at 15 mA:

(2) Kinetic reaction orders of TFA and cobalt catalyst at 10 mA:

As demonstrated above, both TFA and the cobalt catalyst maintain first-order kinetics at 15 mA and 10 mA, consistent with the results obtained under standard conditions at 20 mA. These results confirm that changing the current density does not alter the rate-determining step in this reaction. We have included a discussion of it in the Supplementary Fig. 20 and 21.

2. In terms of the reaction pathway, while the outer-sphere mechanism (e.g., TS-2 in Fig. 5f) is compelling, have the authors explored the possibility of an inner-sphere mechanism to rule out alternative pathways?

Our response: We have indeed considered the possibility of the inner-sphere mechanism. Extensive computational efforts to identify precursors or transition states for the inner-sphere pathway from **Int-2** were unsuccessful, as all optimization attempts converged to either **Int-3** or **TS-2**. Furthermore, molecular orbital analysis revealed that the HOMO of **Int-2** is predominantly composed of the d_{z^2} orbital of the cobalt center, which occupies the vacant coordination site and sterically hinders oxygen coordination of trifluoroacetic acid (TFA) from both the top and bottom faces. This orbital configuration rationalizes the unlikelihood of the inner-sphere mechanism. The HOMO orbital analysis of **Int-2** has been included in the Supplementary Fig. 34 to provide additional mechanistic insights.

Additionally, we explored several potential coordination modes where TFA replaces the MeCN ligand (see Supplementary Fig. 24, **Int-6**, **Int-7**, **Int-8**, and **Int-9**). These alternative pathways were found to be significantly less favorable than the outer-sphere pathway leading

to **Int-3** (as depicted in Fig. 5f of the main text), with energy differences exceeding 5 kcal/mol in most cases. These findings collectively confirm that the outer-sphere mechanism is the dominant pathway and that the inner-sphere pathway can be confidently ruled out.

3. In Fig. 5f, a potential intramolecular 1,3-H shift from intermediate Int-2 may also exist as an alternative route to liberate trifluoroacetaldehyde while generating the Co(I)-OH species. This process should be discussed, as this alternative route might offer additional mechanistic insights. Our response: We have conducted additional calculations to explore the potential intramolecular 1,3-H shift from intermediate **Int-2** as an alternative route to liberate trifluoroacetaldehyde and generate the Co(I)-OH species. As shown below, the energy barrier for this process is calculated to be 28.7 kcal/mol, and the formation of the resulting Co(I)-OH species is endothermic by 8.6 kcal/mol relative to **Int-2**. These energetic results indicate that this pathway is kinetically prohibitive and thermodynamically unfavorable compared to the dominant pathway described in the manuscript. Consequently, the 1,3-H shift mechanism is unlikely to play a significant role under the reaction conditions studied. **The detailed results of this investigation have been also incorporated into the Supplementary Fig. 35.**

4. Did the author observe any defluorinative side-reactions with trifluoroacetic acid during the electrocatalytic process in Table 1?

Our response: the corresponding reductive defluorination byproducts were not detected under either the optimized conditions or variations of the standard conditions listed in Table 1. **We mentioned this result in the manuscript page 4, line 76.**

5. Compared to trifluoroacetic acid, significantly lower yields with simple alkyl carboxylic acids were observed in Fig. 2. An explanation for these results is required.

Our response: When the reactions were performed with alkyl carboxylic acids as the substrates, we detected an obvious side product—substituted quinoline, which is generated through a sequence of aldol condensation and Doebner-von Miller reaction (mechanism illustrated below). **We have mentioned this result in the manuscript page 6, line 137 and included a discussion of**

it in the Supplementary Fig. 8.

6. When primary amines were used as substrates, was there any detection of dialkylated products? Clarifying this would provide insight into the reaction's selectivity.

Our response: three carboxylic acid substrates (CF₃COOH, EtCOOH and HCOOH) were evaluated under the electrocatalytic conditions, and we found both CF₃COOH and EtCOOH predominantly yielded their corresponding mono-alkylated products, whereas less sterically hindered HCOOH generated a mixture of mono- and di-alkylated products. We have mentioned this result in the manuscript page 4, line 76.

Entry	Acid substrate	Yield of mono-alkylation	Yield of di-alkylation
1	CF ₃ COOH	91	0
2 ^a	EtCOOH	47	trace
3	HCOOH	23	12

^a Substituted quinoline side product observed as shown above.

7. Regarding the role of Lewis acids such as Ti(OBu)₄ and BF₃·OEt₂ in activating carboxylic acids, could experimental or computational evidence be provided to substantiate the proposed mechanism?

Our response: To investigate the interaction between carboxylic acids and Lewis acids, we conducted ¹³C NMR spectroscopy. As shown in the spectra below, the addition of Ti(OBu)₄ to trifluoroacetic acid (TFA) resulted in downfield shifts of the carbonyl carbon resonance. Similarly, when BF₃·OEt₂ was introduced to propionic acid, observable changes in the chemical shift of the carbonyl group were also detected. These findings demonstrate that Lewis acids effectively activate the carbonyl groups of carboxylic acids. We have mentioned these results

in the manuscript on page 8 lines 194-196 and included a discussion of it in the Supplementary Fig. 30.

Reviewer #2 (Remarks to the Author):

In the titled manuscript, Liu and coworkers reported an interesting electrocatalytic reductive amination of carboxylic acids via cobalt catalysis. This efficient and sustainable method utilizes a key cobalt-hydride intermediate to facilitate the hydrogenation of carboxylic acids with amines, providing a valuable approach for synthesizing various amines and deuterated derivatives. The products are well-characterized, ensuring reproducibility for researchers interested in amine synthesis. Furthermore, control experiments, kinetic isotope effect studies, and computational analyses support the proposed reaction mechanism. Therefore, this manuscript could be published in Nature Commun, provided that the following points are sufficiently addressed.

1, In terms of carboxylic acids scope, trifluoroacetic acid demonstrated high efficiency, achieving a 93% yield of product. However, simple aliphatic carboxylic acids (e.g., acetic acid, propionic acid and others) provided much lower reactivities. Could the authors explain the poor performance of aliphatic carboxylic acids in this system, including any observed side reactions? What happens when aromatic carboxylic acids are used in this reductive reaction?

Our response: In comparison with trifluoroacetic acid, alkyl carboxylic acids exhibit reduced carbonyl electrophilicity and consequently lower reactivity, necessitating more stringent reaction conditions including the employment of stronger Lewis acids (e.g., $\text{BF}_3 \cdot \text{OEt}_2$) for effective carbonyl activation. Furthermore, when employing simple alkyl carboxylic acids as substrates, competitive formation of substituted quinolines was observed through the Doebner–von Miller pathway, as previously illustrated. As a result, these two aspects—diminished intrinsic reactivity and competing side reactions—lead to alkyl carboxylic acids substantially less efficient than trifluoroacetic acid in this electroreductive transformation. Notably, attempts to utilize benzoic acid as a substrate showed no observable conversion of the starting material under the applied conditions. We have mentioned this result in the manuscript page 6, lines 137-139 and included a discussion of it in Supplementary Fig. 8.

2, Are simple amino acids compatible with this electrochemical system, including protected or unprotected amino acids?

Our response: as shown in the result as below, we evaluated several amino acids, including unprotected variants, as potential substrates. However, no desired product formation was observed, with only amidation side products being detected. We have included these results in Supplementary Fig. 8.

3, Regarding the amine substrates, how do alkylamines react with trifluoroacetic acid under electrocatalytic conditions, particularly considering they form products when formic acid is used as the reagent? Furthermore, would this electrochemical system be suitable for processing substituted hydroxylamine derivatives?

Our response: We investigated the electroreductive coupling of trifluoroacetic acid with alkylamines (e.g., benzylamine and morpholine); however, the desired N-alkylation products were not observed. Interestingly, when phenylhydroxylamine was employed as the substrate, an unexpected N-alkylated benzidine derivative was isolated in 35% yield. We propose that this transformation may proceed via a benzidine rearrangement pathway (see below). We have mentioned these results in the manuscript on page 5 lines 112-115 and included these results in Supplementary Fig. 8.

4, The function of $\text{Ti}(\text{OBu})_4$ as an additive in this electrocatalytic reduction needs to be elucidated.

Our response: We have addressed this issue in response to Reviewer 1 - please refer to question 7 of Reviewer 1.

5, What are the possible reaction intermediates when using nitroarenes as substrates? A plausible reaction pathway is required.

Our response: The electrochemical reduction of nitroarenes is a complex multistep process that proceeds through multiple intermediates. For example, nitrobenzene likely undergoes a six-electron reduction pathway to generate potential intermediates including nitrosobenzene,

hydroxylamine, and ultimately aniline. The resulting aniline then participates in reductive amination with the carboxylic acid substrate to afford the desired N-alkylated product, constituting an overall ten-electron transfer process. Additionally, other potential intermediates such as azoxybenzene, azobenzene, and hydrazobenzene may also contribute to alternative reaction pathways. **We mentioned the discussion of it in Supplementary Fig. 4.**

6, Given that the authors identified aldehydes as reduction intermediates in their mechanistic studies, has the possibility of imine intermediates been explored? Were any imine species detected under the reaction conditions?

Our response: Following the reviewer's suggestion, we evaluated the corresponding imines as substrates and successfully obtained the desired alkylated products in yields exceeding 70%. Notably, the imine intermediates were not observed by NMR, presumably due to their rapid reduction to the corresponding amines—outpacing the rate-limiting reduction of the carboxylic acid to the aldehyde. **We mentioned the discussion of it in Supplementary Fig. 9.**

7, Please carefully check the tables, figures and main text, and unify the format of the reaction equations, e.g. some type errors in Fig. 2 and Fig.4.

Our response: these type errors have been corrected.

Reviewer #3 (Remarks to the Author):

Liu, Qu and co-workers developed an electrocatalytic reductive amination strategy that achieved hydrogenative coupling between carboxylic acid compounds and amines/nitroarenes. This research employed protons as the hydrogen source and a cobalt complex as the catalyst, which in situ generated active cobalt-hydride catalytic species under electrochemical conditions, thereby enabling efficient construction of alkylamine compounds. Through controlled experiments including cyclic voltammetry analyses and ¹H NMR experiments, the authors monitored and characterized the Co(I)-H species. Based on controlled experiments including amide/aldehyde additive comparisons and kinetic isotope effect studies, combined with DFT calculations, the authors proposed a reaction mechanism involving the reduction of carboxylic acids to aldehydes followed by reductive amination to construct alkylamine compounds. In conclusion, this study on the selective cobalt electrocatalytic reductive amination of carboxylic acids using protons as the hydrogen source demonstrates sufficient novelty in the reaction design and a thorough exploration of the reaction mechanism. This reviewer believes the manuscript can be published in Nature Communications after a minor revision.

1. In Fig. 5f, this reviewer suggests that the alternative pathway involving direct β-O elimination of Int-2 to generate a Co(I)-OH complex and trifluoroacetaldehyde simultaneously could be considered.

Our response: The possibility of a β-O elimination pathway is calculated. Our calculations indicate that the formation of the Co(I)-OH complex via β-O elimination from Int-2 is endothermic by 8.6 kcal/mol. Importantly, the resulting intermediate **Int-13** lies 21.9 kcal/mol higher in energy than **Int-4**, the key intermediate along the dominant protonation pathway described in the manuscript. This substantial energy difference provides compelling evidence that the β-O elimination pathway is thermodynamically unfavorable and unlikely to compete with the dominant mechanism. Given these findings, we believe that it is unnecessary to pursue the detailed transition state for this pathway. **These calculated results have been included in the Supplementary Fig. 36.**

2. In the Computational Methods section, please provide explicit details regarding the parameter setting of mixed-solvent between Toluene and Acetonitrile to ensure reproducibility and facilitate independent validation by readers.

Our response: We thank the reviewer for this suggestion to enhance the transparency of our computational methodology. In response, we have provided explicit details regarding the parameter settings for the mixed-solvent system composed of toluene and acetonitrile. The solvent parameters were calculated as a weighted average based on a 1:1 volume ratio of toluene to acetonitrile. The specific solvent parameters were defined as follows: $\epsilon_{\text{ps}} = 19.03$, $\epsilon_{\text{psinf}} = 2.02$, $\text{HBondAcidity} = 0.035$, $\text{HBondBasicity} = 0.23$, $\text{SurfaceTensionAtInterface} = 40.725$, and $\text{CarbonAromaticity} = 0.4285$. These values were derived from the Minnesota Solvent Descriptor Database. In the revised manuscript, these solvent parameters have been explicitly detailed within the 'Computational Methods' section (in Supplementary Information page 26) to facilitate independent validation and reproducibility of our computational results.

3. With reference to Fig. 5f of the main text and theoretical study on Co–H catalyzed hydrogenation of ketones (Catal. Sci. Technol., 2019, 9, 5315-5321), the three-coordinate alkylamine cobalt(I) complex generated by migratory insertion of imine into the Co(I)–H bond in Fig. S25 may be unstable and coordinatively unsaturated. Therefore, it is recommended to consider appending another ligand (e.g., acetonitrile) to stabilize the intermediates Int-11 and to promote the subsequent protonation step.

Our response: Further analysis of our computational results revealed that in intermediate **Int-11**, formed via migratory insertion of the imine into the Co(I)–H bond, there is an agostic interaction between the Co center and the C–H bond of the alkylamine ligand. This interaction effectively maintains the four-coordinate nature of the Co center, thereby stabilizing **Int-11** despite its apparent three-coordinate geometry. However, to thoroughly address the reviewer's suggestion, we also explored an alternative pathway in which an additional acetonitrile ligand coordinates to the Co center (Supplementary Fig. 33). Our calculations demonstrated that this

alternative pathway is less favorable than the originally reported mechanism. These findings confirm that the original pathway remains the dominant route. The additional computational results have been included in the revised Supplementary Information to provide a comprehensive understanding of the reaction mechanism. This literature (Catal. Sci. Technol., 2019, 9, 5315-5321) has been also cited in the manuscript as ref. 75.

We sincerely appreciate the constructive comments from the three reviewers, which have significantly improved our manuscript.